# Cytokines Profile and Lung Function in Children with Obesity and Asthma: A Case Control Study

**DOI:** 10.3390/children9101462

**Published:** 2022-09-24

**Authors:** Laura Maffeis, Carlo V. Agostoni, Denise Pires Marafon, Leonardo Terranova, Claudia Giavoli, Gregorio P. Milani, Mara Lelii, Barbara Madini, Paola Marchisio, M. Francesca Patria

**Affiliations:** 1Pediatric Unit, Fondazione IRCCS Ca’ Granda, Ospedale Maggiore Policlinico, 20122 Milan, Italy; 2DISCCO, Università Degli Studi di Milano, 20122 Milan, Italy; 3Section of Hygiene, University Department of Life Sciences and Public Health, Università Cattolica del Sacro Cuore, 00168 Rome, Italy; 4Internal Medicine Department, Fondazione IRCCS Ca’ Granda, Ospedale Maggiore Policlinico, 20122 Milan, Italy; 5Endocrinology Unit, Fondazione IRCCS Ca’ Granda, Ospedale Maggiore Policlinico, 20122 Milan, Italy; 6DEPT, Università degli Studi di Milano, 20122 Milan, Italy

**Keywords:** asthma, pediatric obesity, inflammation, cytokines, lung function

## Abstract

The existence of common inflammatory biomarkers linking obesity and asthma in children has been hypothesized. Nevertheless, laboratory and clinical characteristics of children with obesity and asthma are still poorly defined. The primary aim of the present study is to investigate the lung function and the cytokine profile, in children with obesity and asthma. In this prospective, cross-sectional pilot study, pulmonary function tests, biochemical parameters, and serum cytokines levels were compared in three groups of 28 children each, matched for age and sex. Obese children showed normal forced spirometry values except an increased distal airway resistance in subjects with obesity and no asthma. Both groups including obese children showed higher leptin and IL-10 levels and lower adiponectin and TNF-alpha levels compared to children with no obesity and asthma. IL-33 and TGF-beta1 levels were higher in children with obesity and asthma vs. children with normal weight and asthma. Finally, IL-6 was undetectable in approximately 70% of obese children with no asthma, in 57% obese asthmatic children and in 100% of children with normal-weight and asthma. Children with obesity and asthma show the most striking cytokine profile, suggesting a pro-inflammatory role of fat mass in asthma development.

## 1. Introduction

The global prevalence of both obesity and asthma in childhood has been rapidly increased in the past few decades [1]. Obesity and asthma are chronic diseases of multifactorial pathogenesis and many studies in children highlight the probable existence of common risk factors which can explain the association between these two conditions [2,3]. However, the mechanisms underlying the concurrent development of these conditions remain poorly understood. At least for childhood, many longitudinal studies suggest that overweight/obesity may precede the onset of asthma [4,5]. It is well-known that the fat accumulated in the thoracic and abdominal spaces reduces the compliance of the respiratory system altering the lung function [6]. Moreover, adipose tissue plays an active role in metabolic, neuroendocrine, and immune functions and growing evidence show that fat tissue itself may independently promote inflammation, disrupting the balance of cytokine, chemokine, and leading to airway inflammation [7]. Therefore, a peculiar phenotype of “obesity with asthma” could exist in children, although its clinical and functional characteristic remain poorly defined. While many studies have reported that “obesity with asthma” is more often associated with a Th2 pathway, including elevated IgE and eosinophilic inflammation, other studies documented that obesity is associated to a non-Th2 inflammatory state, triggered by adipocyte hypoxia with release of the pro-inflammatory leptin, which shifts the macrophage pool in visceral fat from M2 to M1 macrophages. In turn, M1 macrophages start a cascade of pro-inflammatory cytokines such as IFN-γ, IL-6, TNF-α, IL-1β, and monocyte chemotactic peptide (MCP)-1 [8,9]. Adipokines and other cytokines, such as IL-6, secreted by adipose tissue, can also affect the airways through systemic inflammation and metabolic dysfunction [10]. An early recognition of the “obesity with asthma” phenotype would be crucial for optimizing the treatment because obese children seem to have increased asthma severity and reduced response to asthma medications [11].

In order to better identify the “obesity with asthma” specific phenotype, the primary aim of this preliminary study was to assess the lung function and the cytokines profile in a group of children with obesity and asthma. Furthermore, we aimed to compare the cytokines profile of children with obesity and asthma with those of a group of children without obesity and with asthma and of a group of children with obesity and without asthma.

## 2. Materials and Methods

This single-center case-control study was performed at the outpatient clinics for pediatric respiratory disease and pediatric nutrition of the Fondazione IRCCS Ca’ Granda Ospedale Maggiore Policlinico, Milan between January and December 2018. The following three groups of subjects were eligible for the study: (1) children with abnormally high body mass index and asthma (O + A), (2) children with abnormally high body mass index without asthma (O + NA), and (3) children with a normal body mass index and asthma (NO + A). Subjects of the three groups were matched for age (±1 year) and sex. Children with underlying systemic chronic diseases (e.g., children with a hemodynamically significant heart disease) were excluded. Furthermore, we ruled out children with an intercurrent illness (e.g., children with flu-like symptoms).

Abnormally high body weight (overweight and obesity) was defined according to the WHO reference for children aged 5–19 years [12]. The diagnosis of asthma was performed with a history of respiratory symptoms (wheeze, shortness of breath, chest tightness, and cough) and evidence of variable expiratory flow limitation (FEV1/FVC ratio < 0.85) in at least one spirometric evaluation, according to the Global Initiative for Asthma [13].

At the enrollment, after signing the informed consent form, parents filled-in a standardized, self-administered questionnaire about anamnestic and clinical data of the child. Pulmonary function tests were carried out, at least 30 days after discontinuation of inhaled therapy, through the HypAir Compaq spirometer (Medisoft spirometer, Belgium). Forced spirometry (FEV_1_, FVC, FEV_1_/FVC, MEF_25–75_), measurement of distal airway resistance (R_OSC 5Hz–20Hz_) by oscillation technique (R_osc_), residual volume (RV) with multiple breath helium dilution method, and diffusion capacity for carbon monoxide (DL_CO_) were performed according to the ATS/ERS recommendations [14,15,16]. The fraction of exhaled nitric oxide (FeNO) (NIOX VERO^®^, Circassia, Sweden) was also measured for each subject enrolled. The spirometric results were expressed according to the Global Lung Function Initiative (GLI 2012) reference values.

Venous whole blood samples were drawn on the same day of pulmonary function tests. Eosinophil count, total IgE, total cholesterol, triglycerides, glycaemia, ferritin, 25-OH-vitamin D, vitamin A, and vitamins E levels were assessed. Moreover, the levels of adiponectin, leptin, IL-6, IL-10, IL-33, TNF-alpha, and TGF-beta1 evaluation, were measured through enzyme linked immunosorbent assay method (ELISA, Cusabio USA).

Results of cytokine levels were calculated by means of Curve Expert Software (v1.4, MyBioSource.com, San Diego, CA, USA) using four parameters logistic (4-PL) curve fit and were expressed in nanograms/milliliter (ng/mL) for adiponectin, leptin, and TGF-beta1 or picograms/milliliter (pg/mL) for IL-6, IL-10, IL-33, and TNF-alpha.

The study protocol and the collection of patient’s information were approved by the Ethics Committee of the Fondazione IRCCS Ca’ Granda Ospedale Maggiore Policlinico (Ethic Committee, Milan, Area 2, approval number 333_2016). Parents or legal guardians of the enrolled subjects signed an informed consent.

Continuous variables, expressed as medians, interquartile ranges [IQR], and minimum-maximum values, were compared using the Mann–Whitney U test. Categorical data, expressed as absolute frequencies and percentages, were compared among groups by means of the Chi-square test, or the Fisher exact test, as appropriate. Correlation between quantitative variables was evaluated using Spearman’s rank coefficient (rho). All statistical tests were two-sided; a *p* value < 0.05 was considered statistically significant. Statistical analyses and graphs were performed with GraphPad Prism 5.0 (GraphPad Software, Inc., San Diego, CA, USA). Due to the explorative purpose of our preliminary survey, 28 subjects per group had to be enrolled under the hypothesis that the prevalence of subjects with undetectable IL-6 (values < 0.2 pg/mL) was 60% and 20% among overweight/obese children with and without asthma, respectively, with a 5% significance level (two-sided) and a power of 80%.

## 3. Results

Three potentially eligible subjects in the O + A group and Two subjects in O + NA group declined to participate. Finally, we included 28 consecutive children in the O + A, 28 in the O + NA and 28 in the NO + A group. In the O + A group and in O + NA four subjects were overweight and 24 were obese. The demographic and clinical features of the enrolled subjects are reported in Table 1.

The three groups were similar regarding gestational age, birth weight, presence of exclusive breastfeeding, exposure to passive smoking during pregnancy and after birth. BMI was slightly higher in obese non-asthmatic, compared to obese asthmatic children, while the age of overweight development was comparable in the two groups. The severity of asthma was comparable in the O + A and NO + A groups, as well as the age of asthma onset, the presence of exercise-induced dyspnea and the number of subjects actively taking a controller medication (inhaled corticosteroids with or without long-acting beta_2_ agonists) for at least two years (median 3.4 [2.4–8.5] years vs. 3.9 [2.8–9.6] years in in O + A and NO + A, respectively).

Lower respiratory tract infections were slightly more frequent in the asthmatics, while upper respiratory tract infections were overlapping in the three groups. Table 2 shows the respiratory function of the examined groups.

Obese children (O + A and O + NA) showed normal and overlapping forced spirometric values, while FEV_1_/FVC and MEF_25–75_ were significantly lower in normal-weight asthmatic children as compared to obese non-asthmatics. Respect to the two asthmatic groups, obese non-asthmatic children experienced an increased R_OSC 5Hz–20Hz_, while the RV, despite increased in all the three groups, was significantly higher in normal-weight asthmatics compared to obese asthmatic children. The measurement of DL_CO_ was in the normal range in all three groups. Finally, FeNO, were higher in all the children suffering from asthma compared to obese children with no asthma.

In the two groups with obese children, an inverse correlation, despite within a normal range, was found between the age of onset of obesity and respiratory function indices (FEV_1_, MEF_25–75_) (Figure 1).

Peripheral blood eosinophils and total IgE levels were higher in asthmatic children (O + A and NO + A) and were consistent with the more frequent allergic sensitization in these two groups (Table 3).

Obese children showed significantly higher total cholesterol and triglycerides values, while no differences were found between the three groups regarding glycemia, ferritin (even if with a higher trend in groups with obesity), and fat-soluble vitamins levels.

Concerning the serum cytokine pattern, leptin and IL-10 levels were significantly higher in all obese children. Adiponectin and TNF-alpha were, instead, significantly higher in asthmatic, normal weight children compared to the obese subjects. IL-33 and TGF-beta1 levels were both higher in children with obesity and asthma, and showed an intermediate value in children with obesity and no asthma and were lower in children with asthma and normal weight (Figure 2).

Finally, IL-6 was undetectable in 19/28 obese children with no asthma (67.8%), in 16/28 obese asthmatic children (57.1%) and in 28/28 children (100%) with normal-weight and asthma.

## 4. Discussion

This preliminary prospective case-control study investigates for the first time the lung function and the cytokines profile in children with and without obesity and asthma.

In this investigation, we documented that obese children showed a normal forced spirometry; a similar spirometric pattern between obese asthmatic and normal-weight asthmatic children was also observed. However, despite asymptomatic, obese non-asthmatic children showed abnormalities in distal lung function with elevated peripheral airways resistance (R_OSC_) and increased residual volume (RV). A characteristic distribution of the serum cytokine pattern has also been documented, since leptin and IL-10 have been found to be increased in obese patients, while adiponectin and TNF α were significantly higher in the normal-weight asthmatic group. Accordingly, IL-33 and TGF β1 increased in obese asthmatic children, with an intermediate value in the obese non-asthmatic group and lower levels in normal-weight asthmatic subjects. Ferritin (a biomarker of the inflammatory potential of fat tissue) showed a milder higher trend in the two groups with obesity, and future studies should take it into account in designing the power of the sample size. In our study, the pre-enrollment screening did not include the assessment of the degree of obesity and BMI was slightly higher in the group with obesity and without asthma. However, our population was homogeneous with respect to other demographic and clinical features and, within this context, we did not find any protective effects of exclusive breastfeeding on the development of obesity. The association of breastfeeding and later weight and fat development has long been discussed and many factors may contribute [17], including the maternal BMI status [18], making any conclusive remark just speculative.

Since an allergic sensitization was significantly less frequent in obese non-asthmatic children, obesity alone seems not to play a facilitating role on eosinophilic allergic inflammation [19]. Studies on adults have shown that obesity is mainly associated with non-atopic asthma and in the U.S. Severe Asthma Research Program (SARP), the authors identified a cluster of patients with severe asthma that were predominantly affected by obesity, had late-onset asthma and relatively low atopy [20]. Conversely, in our study asthmatic subjects (obese or with normal weight) showed high and overlapping prevalence of allergic sensitization, maybe through an epithelial barrier dysfunction and a loss of innate and adaptive immunity imbalance [21]. The limited number of patients of our study does not allow to draw definitive conclusions. In line with several studies, obese non-asthmatic children showed an asymptomatic increase in the more distal airway resistances, not detected by forced spirometry [22,23], but peripheral airways compression, could, over time, lead to a symptomatic reduced lung function, because of a local proinflammatory response related to a continuous damage of the airway epithelium [24].

Obese asthmatic children have normal FEV_1_, FEV_1_/FVC, and MEF_25–75_ values, mostly comparable to those of normal-weight asthmatic patients. These findings are consistent with several studies that reported only a slight reduction of FEV_1_ and FVC in the presence of obesity, with FEV_1_/FVC ratio usually unaffected [25,26]. Nevertheless, most asthmatic patients have long been treated with inhaled steroids that were discontinued 30 days before the enrollment and this treatment could have, at least in part, affected the respiratory function. In all children, residual volume (RV) was higher than normal range, although lower in subjects with obesity compared to the ones with normal weight and asthma. This is in contrast with other studies which documented that obesity has little effect on RV and lung volumes and could be explained by the small sample size and by the degree of obesity, not severe in our population, differently than in severe obesity [27]. All the obese children showed a normal CO diffusing capacity (DL_CO_) too, in contrast with Li et al. who found in 64 children with obesity a diminished DL_CO_, possibly reflecting the structural changes in the lung interstitium, resulting from lipid deposition and/or decreased alveolar surface area [28]. Other authors found normal or slightly increased DL_CO_ values in adult obesity, likely related to an increased pulmonary blood volume [29].

Children with obesity, regardless of the presence of respiratory symptoms, showed an inverse correlation between the time of onset of obesity and respiratory function indices (FEV_1_ and MEF_25–75_). It could be assumed that a long-standing obesity (i.e., obesity started around 1 year of age) may not be recognized as “abnormal” by a developing lung, which would therefore adapt over time to the obesity condition. This hypothesis has been suggested by Zhang et al. in a population of 285 full-term newborns at high-risk of developing asthma, enrolled in the Childhood Origin of Asthma (COAST) project. In a longitudinal analysis, the authors documented that being overweight at 1 year of age was associated with a decreased risk of asthma at age 6 and 8 years, as well as better lung function [30], consistent with the Barker’s hypothesis of the positive programming of higher weight values at birth and 1 year of age [31]. Accordingly, earlier higher weight gains could support anatomical and functional lung development with increased alveolarization [32].

Most cytokine levels were higher in the two groups with obesity (Figure 3).

IL-33, indirectly activating eosinophils, and associated with chronic inflammation and airway remodeling in asthma [33], was found lower in our children with normal weight and asthma. On the contrary, obese asthmatic children showed the highest levels of IL-33 and TGF-beta1, which are among the most important cytokines involved in airway remodeling and in interactions with adaptive T-helper cells [34]. Compared with children with normal weight and asthma, leptin and IL-10 were significantly higher in all the obese children, indirectly excluding a role of these cytokines, in particular leptin, for the asthma–obesity phenotype. It is likely that increased levels of leptin in association with its defective metabolic pathway can in part contribute to inflammatory processes and increased airway hyperreactivity, described in adult with asthma and obesity [35]. TNF-alpha is the only cytokine, among those here described, to show an inverse trend suggesting a possible role in allergic asthma rather than in the asthma–obesity phenotype. Finally, adiponectin seems to have less value in determining the severity of asthma and does not seem to play a relevant role in identifying the obesity-asthma phenotype [36]. In our study IL-6 was mostly undetectable in all the patients, in spite it could be a useful biomarker of systemic inflammation and metabolic dysfunction [37].

Some recent findings have suggested that innate limphoid cells (ILC), mostly ILC2, activated by different cytokines including IL-33, could be also involved in obesity-associated allergic airway inflammation [38]. It is also recognized that IL-33 tends to be high in hypoxic tissues [38,39]. These mechanisms might explain the higher levels of this cytokine observed in obese children included in this study. Similarly, TGF-β production is upregulated in obesity and, through its profibrotic and proinflammatory functions, this cytokine seems to play a key role in obesity-associated inflammatory airway diseases such as asthma [40]. Our findings confirm this hypothesis. However, due to the small sample size, such considerations remain mainly speculative.

The present study is subject to a number of potential methodological weaknesses. First, the small sample size may limit the generalizability of the findings. Second, overweight and obesity are defined as BMI and not as a percentage of fat mass or as the real distribution of body fat. Although BMI shows a high degree of correlation with the percentage of body fat, people with the same BMI may have very different body composition, which may result in metabolic and lung function differences. Third, patients with asthma, either with obesity or with normal weight have been treated with asthma controller medication for a variable amount of time before the enrollment and this could have modified the objective measurement of the lung function and the related inflammatory pattern. Fourth, in the group of obese children with no asthma, the evaluation of dyspnea was made on anamnestic data, leading to a possible underestimation of the disease. Fifth, we did not collect data from non-asthmatic, normal body weight children. Finally, we did not analyze all blood parameters that are potentially associated to the inflammatory process (e.g., IL-1β).

## 5. Conclusions

In conclusion, in this preliminary case-control study we documented that children with obesity showed a normal forced spirometry, as well as spirometry was overlapping between children with obesity and asthma and children with normal weight and asthma. Even if asymptomatic, obese children with obesity with no asthma showed slight abnormalities in distal lung function with elevated peripheral airways resistance (R_OSC_) and increased residual volume (RV). A characteristic distribution of the serum cytokine pattern has been documented with prevailing higher levels in children with obesity and, more markedly, in patients with obesity and asthma. These findings seem to confirm the pro-inflammatory role of fat mass in children with asthma. On the other hand, the long-term predictivity of asthma development in older ages should be still explored in larger sample size observations to elucidate the clinical profile of the asthma–obesity phenotype.

## Figures and Tables

**Figure 1 children-09-01462-f001:**
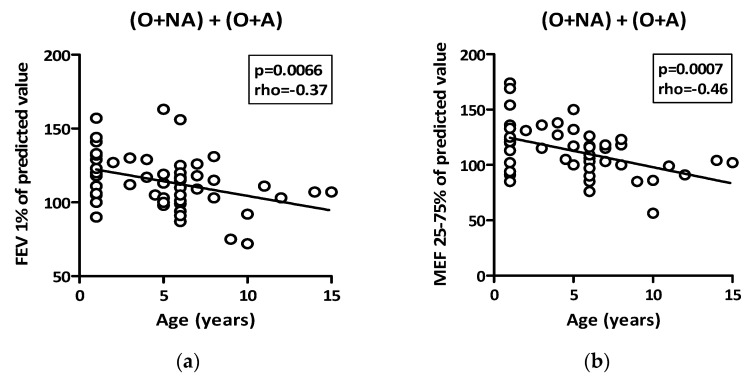
Correlation between age of onset of obesity and respiratory function indices (rho = Spearman’s rank correlation coefficient). (**a**) Correlation between age and FEV1%; (**b**) Correlation between age and MEF 25–75%.

**Figure 2 children-09-01462-f002:**
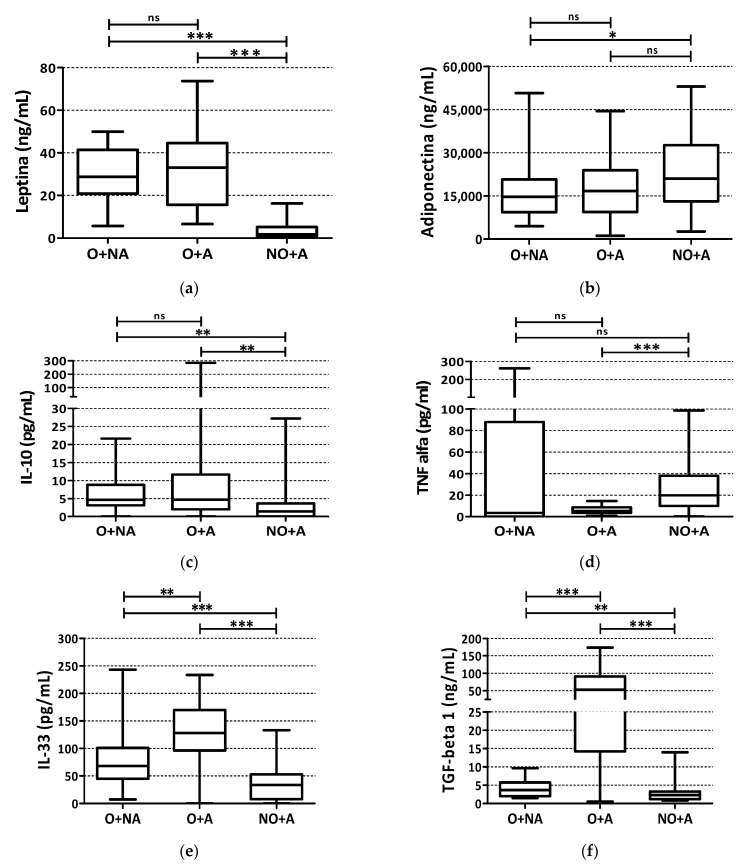
Cytokine pattern of children with abnormally high body mass index without asthma (O + NA), with abnormally high body mass and asthma (O + A) and children with asthma and normal body weight (NO + A). The boxes include the interquartile values, horizontal line in the box points out median value, the whiskers include minimum and maximum values. Statistical test: Mann–Whitney *U* test (* *p* < 0.05; ** *p* < 0.01; *** *p* < 0.001; ns = *p*-value not significant). (**a**) Leptina (ng/mL); (**b**) Adiponectina (ng/mL); (**c**) IL-10 (pg/mL); (**d**) TNF-alfa (pg/mL); (**e**) IL-33 (pg/mL); (**f**) TGF-beta 1 (ng/mL).

**Figure 3 children-09-01462-f003:**
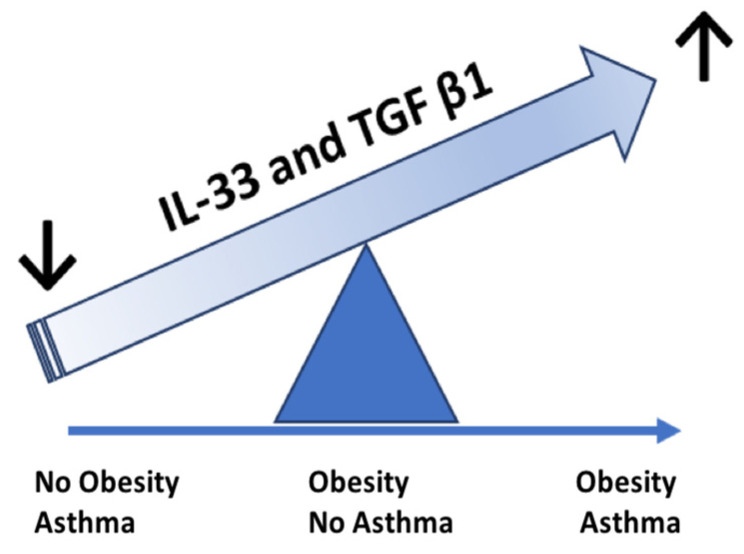
Hypothesis on the role of IL-33 and TGF β1 in the transition from non-asthmatic obese phenotype to obese asthmatic phenotype. Based on the IL-33 and TGF β1 levels, the child with obesity and no asthma is placed between the subject with normal weight and asthma and the subject with obesity and clinical asthma. With the increase of IL-33 and TGF β1, the balance progressively moves toward the asthma–obesity phenotype.

**Table 1 children-09-01462-t001:** Patient’s demographic and clinical features. Twenty-eight children with obesity and asthma (O + A) were compared with 28 children with obesity and no asthma (O + NA) and with 28 children with asthma and no obesity (NO + A) overlapped for age (±1 year) and gender (males 71.4%). Values are expressed as median and minimum-maximum (^1^) or absolute value and percentage (^2^). LRTI = lower respiratory tract infections, URTI = upper respiratory tract infections, * = *p* < 0.05, ** *p* < 0.005, *** = *p* < 0.0001.

Demographic and Clinical Features	O + NA	O + A	NO + A	*p*-Value		
				O + NA vs. O + A	O + NA vs. NO + A	O + A vs. NO + A
Age, years ^1^	11.7 (6.8–17.8)	11.8 (6.9–16.7)	11.4. (7.9–16.8)	0.77	0.68	0.89
Sex, male	20 (71)	20 (71)	20 (71)	1.0	1.0	1.0
Gestational age, weeks ^1^	40 (36–41)	40 (34–42)	40 (35–41)	0.65	0.42	0.24
Birth weight, Kg ^1^	3.3 (2.2–4.9)	3.5 (2.0–4.1)	3.4 (2.6–4.2)	0.83	0.83	0.46
Exclusive breastfeeding for at least 3 months ^2^	24 (86)	20 (71)	24 (86)	0.33	1.0	0.33
Parental smoking during pregnancy ^2^	1 (3.6)	3 (11)	2 (7.1)	0.61	1.0	1.0
Exposure to passive smoke after birth ^2^	7 (25)	8 (29)	9 (32)	0.76	0.55	0.77
BMI, Kg/squared m ^1^	27.7 (22.6–38.7)	25.6 (19.8–41.8)	18.1 (14.5–20.6)	0.027 *	<0.0001 ***	<0.0001 ***
Onset of overweight/obesity, years ^1^	5 (<1–14)	6 (<1–15)	NA	0.79	NA	NA
Maternal obesity ^2^	17 (60.7)	11 (39.2)	3 (10.7)	0.09	<0.0001 ***	<0.0001 ***
Onset of asthma, years ^1^	NA	6 (2–14)	5 (1–11)	NA	NA	0.30
Type/Frequency of asthma: ^2^						
Episodic	NA	21 (75.0)	25 (89)	NA	NA	0.16
Persistent	NA	7 (25.0)	3 (10.7)	NA	NA	0.16
Exercise-induced asthma ^2^	NA	19 (68)	13 (46)	0.11	1.0	0.11
LRTI in the last year ^2^	4.0 (14)	11 (40)	13 (46)	0.07	0.019 *	0.59
URTI in the last year ^2^	17 (61)	21 (75)	16 (57)	0.25	0.79	0.16
Allergic sensitization ^2^	12 (43)	25 (89)	25 (89)	0.0005 **	0.0005 **	1.0

**Table 2 children-09-01462-t002:** Lung function testing in the study population. Values are expressed as median and minimum-maximum. FEV_1_: forced expiratory volume in 1 s; FVC: forced vital capacity; MEF_25–75_: mean flow between 25% and 75% of FVC; FeNO: fractional exhaled nitric oxide; RINT: resistance by interrupter technique; R_OSC 5Hz–20Hz_: difference in impulse oscillometry between airways resistance at 5 Hz and 20 Hz; RV: residual volume; CO: carbon monoxide. * = *p* < 0.05, ** *p* < 0.005, *** = *p* < 0.0001.

Lung Function Parameters	O + NA	O + A	NO + A	*p*-Value		
				O + NA vs. O + A	O + NA vs. NO + A	O + A vs. NO + A
FEV_1_ (%)	115 (75–189)	115 (72–163)	111 (84–149)	0.45	0.59	0.67
FVC (%)	120 (109–134)	123 (109–132)	126 (116–139)	0.82	0.31	0.12
FEV1/FVC	85 (80–89)	84 (63–100)	79 (66–91)	0.30	0.0083 *	0.23
MEF_25–75_ (%)	117 (103–126)	106 (56–169)	104 (79–138)	0.12	0.018 *	0.69
FeNO (ppb)	16 (12–21)	28 (6–273)	44 (8–117)	0.0008 **	<0.0001 ***	0.27
RINT (%)	187 (112–501)	186 (111–392)	161 (179–266)	0.78	0.030 *	0.039 **
R_OSC 5Hz–20Hz_ (mBar/L/s)	11.1 (5.4–29)	7.0 (4.8–17.6)	7.8 (5.5–13.2)	0.011	0.009 **	0.27
RV He (%)	214 (168–298)	209 (89–383)	272 (137–448)	0.69	0.08	0.0046 *
Diffusion CO cor (mL/mmHg/min)	25 (15–110)	26 (13–60)	26 (14–41)	0.45	0.65	0.66

**Table 3 children-09-01462-t003:** Biochemical profile of the study population. * = *p* < 0.05, ** *p* < 0.005.

Blood Parameters	O + NA	O + A	NO + A	*p*-Value		
				O + NA vs. O + A	O + NA vs. NO + A	O + A vs. NO + A
Eosinophils (%)	3.4 (1.5–14.5)	5.9 (1.4–17.3)	6.3 (0.5–24.7)	0.011 *	0.013 *	1.0
Total IgE (KuA/L)	149 (3–3334)	514 (23–3018)	743 (48–2800)	0.031 *	0.015 *	0.94
Total cholesterol (mg/dL)	159 (111–209)	163 (95–231)	139 (97–195)	0.38	0.012 *	0.0008 **
Triglycerides (mg/dL)	98 (12–21)	98 (45–304)	68 (26–159)	0.44	0.0024 **	0.028 *
Ferritin, ng/dL	60.3 (21–163)	62.5 (23–133)	52.2 (20–156)	0.75	0.29	0.15
Glycemia (mg/dL)	85 (64–107)	82 (67–112)	90 (55–119)	0.71	0.09	0.051
25 OH vitamin D (μg/L)	19.9 (6.8–69.1)	24.4 (9.0–35.3)	22 (8.0–35.0)	0.20	0.73	0.20
Vitamin E (mg/L)	9.7 (5.1–15.2)	9.8 (5.5–15.9)	10.0 (6.6–15.0)	0.95	0.72	0.63
Vitamin A (mg/L)	0.33 (0.29–0.35)	0.30 (0.16–0.50)	0.32 (0.22–0.41)	0.72	0.45	0.93

## Data Availability

The data presented in this study are available on request from the corresponding author. Due to the protection of personal data, the data is not publicly available.

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
