# Peer review of "Cytokines Profile and Lung Function in Children with Obesity and Asthma: A Case Control Study"

_children, 2022, doi:10.3390/children9101462_

Round 1
Reviewer 1 Report
1. Introduction
I consider this part of the article too short, some information about the role of cytokines, obesity is needed at the current moment of knowledge.
You should add the aim of the study in a paragraph in this section! You changed the aim of the study, from abstract to phenotype identification in introduction.
2. M& Methods
Line 71, how many were excluded?
You should explain better which were your criteria of include/exclude patients.
In this section, please explain how obese patient were chosen.
3. Please insert a table with serum cytokine level comparison between groups.
If you can provide results of the cytokine level between obese groups with and without asthma? Is a correlation there?
4. Please provide information about Asthma and treatment. How about age, gender, years of treatment (asthma)? Could you correlate your results?
As a result, in a state of chronic inflammation, with overproduction of inflammatory mediators, we are facing a cytokine variation. How is it influenced in practice? Could you provide a cytokine profile for each group so we can compare?
5. Does your conclusion show us the purpose of your study and if it was achieved? how can we show that these correlations are useful in practice? Have you followed the effects of varying the cytokine profile to improve the state of knowledge? what conclusions do we add regarding the level of cytokines and diagnostic?
Good luck with your manuscript.

Author Response
- Introduction
- I consider this part of the article too short, some information about the role of cytokines, obesity is needed at the current moment of knowledge.
Authors answer: Thank you for the suggestion. We added some further information about inflammatory cytokines linking obesity and asthma in children in the revised version of the Introduction.
- You should add the aim of the study in a paragraph in this section
Authors answer: This request that was addressed in the revised version of the manuscript
You changed the aim of the study, from abstract to phenotype identification in introduction
Authors answer: we thank the reviewer for this comment. In the revised version of the introduction we clearly specify the primary aim of the study and the other aims.
- M& Methods
- Line 71, how many were excluded?
Authors answer: Thank you for the observation. The information regarding excluded subjects is now reported in the first paragraph of the revised version of results.
- You should explain better which were your criteria of include/exclude patients.
Authors answer: We thank again the reviewer for this constructive comment. The inclusion and exclusion criteria are now clearly reported within the revised version of the methods.
- In this section, please explain how obese patient were chosen
Authors answer: This comment is pertinent. All subjects were consecutively enrolled at the outpatient nutritional clinics of the hospital. This information is now reported within the manuscript.
- Please insert a table with serum cytokine level comparison between groups.
Authors answer: Thank you for this request. The comparison about the three groups is reported in figure 2. Such figure depicts also median, interquartile range, minimum and maximum values and comparison of the three groups. We revised the legend to better clarify such information. We hope that the revised figure reports enough information. If the reviewer still thinks that a table is needed, we are available to add it.
- If you can provide results of the cytokine level between obese groups with and without asthma? Is a correlation there?
Authors answer: The information regarding the comparison between the two groups is given in the revised version of the manuscript (Figure 2).
- Please provide information about Asthma and treatment. How about age, gender, years of treatment (asthma)? Could you correlate your results?
Authors answer: Thanks for your comment. The three groups were overlapped for age and gender, as reported in the Material and Method section and in the attached Table 1. We did not find any difference in the length of controller therapy between the two asthmatic groups (O+A and NO+A). We added this information in the text.
As a result, in a state of chronic inflammation, with overproduction of inflammatory mediators, we are facing a cytokine variation. How is it influenced in practice? Could you provide a cytokine profile for each group so we can compare? Does your conclusion show us the purpose of your study and if it was achieved? how can we show that these correlations are useful in practice? Have you followed the effects of varying the cytokine profile to improve the state of knowledge? what conclusions do we add regarding the level of cytokines and diagnostic?
Authors answer: we really thank the reviewer for this question. Although the present is a pilot investigation conducted on a small local sample size, we found some intriguing differences in the cytokine profile among the three considered groups (leptin and IL-10 levels significantly higher in all obese children; adiponectin and TNF-alpha significantly higher in asthmatic, normal weight children; IL-33 and TGF-beta1 levels were both higher in children with obesity and asthma, had an intermediate value in children with obesity and no asthma and were lower in children with asthma and normal weight; IL-6 was undetectable in many patients of the three groups. However, due to the small sample size, out results cannot be generalized or become a therapeutic target. So far, our findings seem simply to confirm the pro-inflammatory role of fat mass in children with asthma, but many other studies on larger sample size are needed to elucidate the complex clinical profile of the asthma-obesity phenotype and the prognostic value of cytokines and their correlation to therapy.
Reviewer 2 Report
Maffeis et al submitted an interesting article evaluating several cytokine parameters and lung function in children with obesity and asthma (O+A) and compared them with two other groups, children with obesity and no asthma (O+NA) and children with normal weight and asthma (NO+A). Overall, the manuscript was well-written and organized. However, I would suggest some points that might improve the manuscript.
1- If available, data for non-obese non-asthmatic children as another control might support the study further.
2- Line 55-59 (aim of the study): This sentence is very important, so it would be suggested to be clearly written.
3- Line 64 and 68: Links can be replaced by numbers.
4- Table 1: I would suggest using (*) to indicate the statistical significance rather than the bold and add the legend beneath the table.
5- Table 2: I might be wrong but could not find the data for ferritin.
6- The discussion was well-structured, but some suggested reasons regarding the results of cytokines (IL-33, TGF-b1) in NO+A should be added to convince the reader. For example, why NO+A showed a significant lower IL33 and TGF-β1 compared to O+A and O+NA?
7- The limitation paragraph is perfect, but I would suggest adding other points of limitation regarding the blood parameters, such as neutrophils (%), macrophages (%), etc. and a cytokine, such as IL-1β can be used as a pro-inflammatory marker along with TNF.
Author Response
Comments
Overall, the manuscript was well-written and organized. However, I would suggest some points that might improve the manuscript.
If available, data for non-obese non-asthmatic children as another control might support the study further.
Authors answer: Thank you for the relevant observation. The target of our study was the definition of a specific asthma-obesity phenotype, in terms of lung function and cytokine profile; so, in order to better define the target, we thought to investigate three groups (obese, obese with asthma and asthmatic without obesity) in which each group was, in turno, a control group for the others. For this reason we did not foresee a control group of non-obese non-asthmatic group. We acknowledge such point among the limitations of the revised manuscript.
Line 55-59 (aim of the study): This sentence is very important, so it would be suggested to be clearly written.
Authors answer: Thanks for this constructive suggestion. We added the aim of the study in a new paragraph as suggested by the reviewer.
Line 64 and 68: Links can be replaced by numbers.
Authors answer: We modified the manuscript as requested.
Table 1: I would suggest using (*) to indicate the statistical significance rather than the bold and add the legend beneath the table.
Authors answer: We deleted the bold and insert the symbol as suggested by the reviewer
Table 2: I might be wrong but could not find the data for ferritin.
Authors answer: Thanks for the observation. We are sorry for missing ferritin values. We added these data in Table 3.
The discussion was well-structured, but some suggested reasons regarding the results of cytokines (IL-33, TGF-b1) in NO+A should be added to convince the reader. For example, why NO+A showed a significant lower IL33 and TGF-β1 compared to O+A and O+NA?
Authors answer: Thanks for these interesting questions. We addressed this question in the revised manuscript.
The limitation paragraph is perfect, but I would suggest adding other points of limitation regarding the blood parameters, such as neutrophils (%), macrophages (%), etc. and a cytokine, such as IL-1β can be used as a pro-inflammatory marker along with TNF.
Authors answer: We agree with the reviewer. We added the suggested limitation in the revised manuscript.
Round 2
Reviewer 1 Report
I am glad for the way we interacted and now things are better explained. We need studies on larger groups of patients, but your results come to complete what is known at this moment. from my point of view, it is ok now.
Reviewer 2 Report
The manuscript has been greatly improved and the authors have addressed all comments.